# Salt stress induces endoplasmic reticulum stress-responsive genes in a grapevine rootstock

Birsen Çakır Aydemir[1], Canan Yüksel Özmen[2], Umut Kibar[3], Filiz Mutaf[2], Pelin Burcu Büyük[2], Melike Bakır[4], Ali Ergül[2]*

1 Faculty of Agriculture, Department of Horticulture, Ege University, Izmir, Turkey, 2 Biotechnology Institute, Ankara University, Ankara, Turkey, 3 Republic of Turkey, Ministry of Agriculture and Forestry, Agriculture and Rural Development Support Institution, Ankara, Turkey, 4 Faculty of Agriculture, Department of Agricultural Biotechnology, Erciyes University, Kayseri, Turkey

* ergul@agri.ankara.edu.tr

**Citation:** Çakır Aydemir B, Yüksel Özmen C, Kibar U, Mutaf F, Büyük PB, Bakır M, et al. (2020) Salt stress induces endoplasmic reticulum stress-responsive genes in a grapevine rootstock. PLoS ONE 15(7): e0236424. https://doi.org/10.1371/journal.pone.0236424

**Data Availability Statement:** All microarray data (cell files, metadata and matrix Template) was submitted to Gene Expression Omnibus database (http://www.ncbi.nlm.nih.gov/projects/geo/) (GEO

## Abstract

Grapevines, although adapted to occasional drought or salt stress, are relatively sensitive to growth- and yield-limiting salinity stress. To understand the molecular mechanisms of salt tolerance and endoplasmic reticulum (ER) stress and identify genes commonly regulated by both stresses in grapevine, we investigated transcript profiles in leaves of the salt-tolerant grapevine rootstock 1616C under salt- and ER-stress. Among 1643 differentially expressed transcripts at 6 h post-treatment in leaves, 29 were unique to ER stress, 378 were unique to salt stress, and 16 were common to both stresses. At 24 h post-treatment, 243 transcripts were unique to ER stress, 1150 were unique to salt stress, and 168 were common to both stresses. GO term analysis identified genes in categories including 'oxidative stress', 'protein folding', 'transmembrane transport', 'protein phosphorylation', 'lipid transport', 'proteolysis', 'photosynthesis', and 'regulation of transcription'. The expression of genes encoding transporters, transcription factors, and proteins involved in hormone biosynthesis increased in response to both ER and salt stresses. KEGG pathway analysis of differentially expressed genes for both ER and salt stress were divided into four main categories including; carbohydrate metabolism, amino acid metabolism, signal transduction and lipid metabolism. Differential expression of several genes was confirmed by qRT-PCR analysis, which validated our microarray results. We identified transcripts for genes that might be involved in salt tolerance and also many genes differentially expressed under both ER and salt stresses. Our results could provide new insights into the mechanisms of salt tolerance and ER stress in plants and should be useful for genetic improvement of salt tolerance in grapevine.

## Introduction

Salinity is one of the environmental stresses most limiting to crop productivity worldwide [1, 2]. Salinity stress results changes in many physiological and metabolic processes. The extent of these changes varies according to the intensity and length of stress exposed [3–5]. Under salt

ID: GSE150581) using the NCBL (National Center for Biotechnology Information) web server.

**Funding:** Funded by the Scientific and Technological Research Council of Turkey (TÜBİTAK), grant No: 110 O105. The funders had no role in study design, data collection and analysis, decision to publish, or preparation of the manuscript.

**Competing interests:** The authors have declared that no competing interests exist.

**Abbreviations:** DREB, dehydration-responsive element binding; ER, endoplasmic reticulum; ERAD, ER-associated degradation; PCD, programmed cell death; TFs, transcription factors; UPR, unfolded protein response.

stress, reactive oxygen species (ROS) accumulate, cell division is inhibited, and the cellular membrane becomes disorganized [6]. Grapevines are relatively more tolerant to drought stress than to salt stress [7], and are reportedly more sensitive to Cl$^-$ than Na$^+$ toxicity [8]. In grape production, salt stress can be avoided by using salt-tolerant grape rootstocks obtained from wild *Vitis* species [9].

Many studies have examined molecular aspects of the salt stress response in grapevine [10–13]. Transcriptome analysis of the response of *V. vinifera* cv. Cabernet Sauvignon to salt stress [10, 11] showed that transcript profiles among salt and osmotic or water-deficit stresses overlap. In grapevine, perception of salt stress sets signaling pathways in motion and activates transcription factors (TFs) that control stress-responsive genes.

The endoplasmic reticulum (ER) is an essential organelle in protein synthesis, maturation, and secretion in eukaryotic cells [14]. After translation in the ribosome, newly synthesized proteins then mature in the ER lumen to function properly. If unfolded or misfolded proteins accumulate in the ER lumen, the cells face ER stress. ER stress then induces the unfolded protein response (UPR), which relieves ER stress [15, 16]. ER homeostasis and protein synthesis is reestablished by the UPR through (i) initiation of expression of chaperones and foldases for promotion of protein folding, (ii) attenuating translation, and (iii) removing unfolded proteins for degradation by the proteasome [15]. If these processes cannot restore cell homeostasis, programmed cell death (PCD) is then induced. The mechanism of the ER stress response is conserved in all eukaryotic cells.

Plants can also exhibit the ER stress response [17–19]. ER can be heat-, cold-, salt-, or drought-stress induced in plants [15, 20]. The spliced form of *bZIP60* (bZIP-type TF) enters the nucleus and activates UPR target genes [21]. *Arabidopsis* plants overexpressing *bZIP60* showed more tolerance to salt stress [22], whereas single mutants in the *slp* (stomatin-like proteins) or *bZip17* genes were sensitive to salt stress. These results suggest that UPR is involved in the salt stress response [15]. Similarly, heat can induce the translocation of bZIP17 and bZIP28 to the nucleus and the splicing of *bZIP60* mRNA [23].

ER-associated degradation (ERAD) is a ubiquitin/proteasome process associated with the ER that degrades misfolded and unfolded proteins in the ER. ERAD can be divided into four steps: (i) recognition, (ii) ubiquitination, (iii) delocalization, (iiii) and then degradation by the 26S proteasome. Based on the location of the misfolded proteins, there are three different ERAD pathways: ERAD-L (lesion in the luminal region), ERAD-C (defect in the cytoplasmic region), and ERAD-M (defect in the ER membrane) [24, 25]. The molecular components of ERAD including homologs of Hrd1/Hrd3/Sel1L, Derlin-1 EDEMS, and C89 are also found in plant genomes [19, 26, 27]. The *Arabidopsis* ERAD mutants Sel1l/hrd3, hrd1a/hrd1b, and Os9 are less tolerant to salt stress [28, 29]. Tunicamycin blocks glycoprotein synthesis and triggers the UPR. These mutants accumulate misfolded proteins and induce the UPR in response to tunicamycin [19, 29–32]. The ERAD complex can alleviate ER stress by removing misfolded proteins, but in the absence of a functional ERAD pathway, the survival and growth of plants is impaired [28, 33]. Another study identified the ubiquitin conjugating enzyme UBC32 as another component of ERAD [34]. UBC32 is homologous to yeast UBC6 and is thought to be a component of the plant DOA10 complex [34]. Interestingly, *ubc32* mutants are more tolerant to salt stress, and overexpression of UBC32 results in increased salt sensitivity, suggesting a linkage between ERAD and salt stress [34]. Even though the above findings suggest a relationship between ERAD, UPR, and salt stress, the molecular mechanisms underlying this relationship have not been explored in plants. Grapevine might also express a conserved ER stress response that might result from salt stress.

In the present investigation, we report the transcriptome responses of leaves of a salt-tolerant grapevine rootstock, 1616C (*V. solonis* x *V. riparia*), to salt stress and ER stress in order to

identify genes with altered expression in response to both stresses. The expression patterns of some of the regulated genes were compared to identify a simultaneous response to both ER and salt stresses at the transcriptional level. To our knowledge, this is the first report to provide insights into the molecular mechanisms of salt and ER stress in a salt-tolerant grapevine rootstock.

## Materials and methods

### Plant material and stress treatments

Grapevine rootstock 1616C (*V. solonis* x *V. riparia*) was obtained from the Agricultural Research Extension unit at Ankara University in Turkey. After removing the leaves from each rootstock, shoots were pruned to one active bud and one node and then surface sterilized in 70% EtOH followed by 10% bleach, and then were cultured on Murashige & Skoog (MS) medium containing 2% sucrose (w/v) according to the methods of [35]. Plantlets were grown in a growth chamber with a 16-h light/8-h dark cycle for 10 weeks at 25 ˚C. Plantlets with 4–5 leaves and 2–4 roots were chosen for use in stress treatments. Experiments were conducted with three treatment groups: the control (C, without any chemical treatment), TM (treated with tunicamycin only), and salt (treated with NaCl only).

Plantlets were subjected to ER stress treatment by immersing their roots in liquid MS medium containing 5 µg mL$^{-1}$ tunicamycin (TM) as an ER stress-inducing reagent at indicated times at room temperature.

Plantlets were subjected to salt treatment by immersing their roots in liquid MS medium containing 400 mM NaCl and 12 mM CaCl$_2$. As revealed by preliminary experiments, a quantity of 400 mM salt was sufficient for decreasing stem water potentials while not harming the plants exposed to salt stress for the indicated times. Ca$^{2+}$ was added to the MS media to prevent a salt-induced Ca$^{2+}$ deficiency [36, 37]. As a negative control, untreated plantlets were grown in liquid MS medium without any chemical. All experiments were repeated in triplicate. This study does not involve the collection of plant from a natural setting.

### Leaf Stem water potential measurements

Measurements of leaf water potentials (in MPa) were performed using a Model 600 pressure chamber (PMS Instrument Company, Albany, USA) following the manufacturer's recommendations. Measurements were performed for ER and salt stressed samples and controls at indicated time points and were repeated three times. Results were statistically analyzed using *t*-tests ($p < 0.05$). Leaf samples were stored at -80 ˚C until RNA isolation.

### RNA extraction

We isolated total RNA from all 1616C leaf samples following the protocol in [38] followed by an additional purification using an RNeasy Mini Cleanup Kit (Qiagen, Valencia, CA). One gram of ground leaf tissue was homogenized in a buffer containing 200 mM Tris-HCl, pH = 8.5, 1.5% (w/v) lithium dodecyl sulfate, 300 mM LiCl, 10 mM sodium EDTA, 1% w/v sodium deoxycholate, and 1% v/v NP-40. Following autoclaving, 2 mM aurintricarboxylic acid, 20 mM dithiotheitol (DTT), 10 mM thiourea, and 2% w/v polyvinylpolypyrrolidone were added immediately before use. Following precipitation with sodium acetate and isopropanol precipitation, samples were extracted once with 25:24:1 phenol:chloroform:isoamyl and then twice with 24:1 chloroform:isoamyl prior to performing LiCl precipitations to remove DNA contamination. RNA was purified using an RNeasy Mini Cleanup Kit (Qiagen, Valencia, CA) according to the manufacturer's instructions. 1.5% formaldehyde agarose gel electrophoresis

was utilized to determine RNA integrity, and Agilent 2100 Bioanalyzer (Santa Clara, CA) was utilized to determine RNA quality using RNA LabChip® assays according to the manufacturer's instructions.

## cDNA synthesis, labeling, and *Vitis* GeneChip® array hybridization

We synthesized cDNA from total RNAs and labeled them following Affymetrix protocols (http://www.affymetrix.com/support/technical/index.affx). Concisely, first-strand cDNA synthesis was performed from 2.5 µg of total RNA, which was followed by second-strand cDNA synthesis by utilizing a 3′ One-cycle cDNA Synthesis Kit (Affymetrix, Santa Clara, CA). Prior to cDNA synthesis, spiking controls (poly-A RNA controls for One-cycle cDNA synthesis) were added to the total RNA. Following cleanup procedure, biotin-labeled antisense cRNA was synthesized using a MEGAscript® IVT Labeling Kit (Affymetrix, Santa Clara, CA), fragmented by $Mg^{2+}$ hydrolysis, and hybridized to GeneChip™ *Vitis vinifera* Genome Arrays (Affymetrix, Santa Clara, CA) for 16 h at 60 rpm, 45 ˚C. The GeneChip® Fluidics Station 450 (Affymetrix, Santa Clara, CA) was utilized for washing the arrays and the GeneChip® 3000 Scanner (Affymetrix, Santa Clara, CA) was used for scanning.

## Quantitative Real-Time PCR (qRT-PCR)

The expression of ER- and salt-stress responsive genes in leaves were confirmed by qRT-PCR analysis. Total RNA was extracted from samples as above and treated with DNase I (Fermentas, USA). First-strand cDNA was performed by using the Transcriptor First Strand cDNA Synthesis Kit (Roche, Switzerland) according to the manufacturer's protocol. Primers used are listed Table 1.

LightCycler® FastStart DNA Master SYBR Green I Kit (Roche, Switzerland) was utilized for preparation of qRT-PCR reactions and reactions were run by using Roche LightCycler 480. Three replicates were conducted to analyze the expression of each gene under each condition. $2^{-\Delta\Delta CT}$ method was used for calculation of relative expression levels [39]. Transcript abundance was normalized to that of eIF4α (*Vitis vinifera* eukaryotic initiation factor 4A-8 (LOC100261822, XM_002277667.3).

**Table 1. List of primers used for qRT-PCR.**

| Affymetrix ID | Gene | Forward primers (5'-3') | Reverse primers (5'-3') |
|---|---|---|---|
| 1611611_at | Auxin-binding protein ABP19a-like | caggtgaacactggtcg | aagttgttgctaaacagcg |
| 1611867_at | Calcium-binding protein CML44-like | ttctatgaatccatttcgaccg | ccctgaaagccttcgcta |
| 1613770_s_at | Protein TIFY 10A-like | cccaaacagctcaaatgacta | cttcccatgccagctaag |
| 1614008_at | Polygalacturonase-like | tgcaaggatgcaccaac | caaacgaaacatgaatttgct |
| 1620319_s_at | R2R3 MYB | tgggaaatcctccggtgaatcg | cccacagcagtcctttagcg |
| 1615147_at | Unknown | gctccagatatgccggacga | gtccgctcaagatctctcgga |
| 1611548_s_at | Unknown | ggcagtgcaaagaacaag | accatcacaatccctatccagttcc |
| 1607193_at | Ubiquinol oxidase 2 | gttaacgtcccggtgatgcg | ccggccattgccattgactac |
| 1609107_at | Zinc finger protein ZAT10 | gaagacccacgagtgctcca | tcccggtggctgtgactaga |
| 1621592_s_at | Dehydrin | atccgaggacgatgggcaag | tacttgtggcgctcgtttgc |
| 1617483_at | RING-H2 finger protein ATL60-like | ggacggtctggaatgtgctg | ggcaggtggagtgggattga |
| 1610721_s_at | Unknown | ggagtcgcagcaaagcaagg | acgagggtctcgaagcagtg |
| 1609744_at | Uncharacterized protein YIR042C | acacatggaggcggatggaa | tctgtggcccagaccatgaa |
| 1608009_s_at | Stilbene syhthase | acgtagctgggatcaatggct | agctgtgccaatggctagga |
| İnternal Control | eIF4α | gatgtgatccaacaggcacaa | catgaaccctcacaccgaga |

## Microarray data processing and analysis

Microarray analysis was performed using BRB-Array Tools Version 4.6.0 (https://brb.nci.nih.gov/BRB-ArrayTools/) [40, 41] and GCRMA was used as the analysis method. Boxplot and histogram graphics were created for all 24 chips. After the GCRMA analysis, 2989 of 16602 probes are selected for further analysis. Probes with expression differences were determined by comparing expression between the values of stress and control samples and among these selected probes, those with more than 2 fold change (up/down) were used for venn diagrams, gene ontology analysis. Gene ontology analyses to assign probes to the domains molecular function, biological process, or cellular components were performed using http://geneontology.org/ web server. Our ontology analysis took into account the fact that a probe could be assigned to more than one knowledge domain in the ontology [42–44]. For pathway analysis, Probe IDs of genes with expression differences were used to download relevant cDNA sequences from the ensembl plant database (https://plants.ensembl.org/biomart/martview). KEGG (Kyoto Encyclopedia of Genes and Genomes) is a database used to understand the functions and utilities of the biological system for each stress (https://www.genome.jp/kaas-bin/kaas_main). All microarray data (cell files, metadata and matrix Template) was submitted to GEO (Gene Expression Omnibus, (http://www.ncbi.nlm.nih.gov/projects/geo/.) database (GEO ID:GSE150581) using NCBL (National Center for Biotechnology Information) web server.

## Results

Leaf Stem water potentials were measured at the time indicated after salinity or TM treatments to optimize stress conditions. Stem water potential decreased gradually up to 24 h after stress treatments (Fig 1). We did not observe any significant differences in stem water potential between salt-stressed and ER-stressed plants at the indicated times. Stem water potentials were almost equal at 24 h, and salt-stressed plants did not wilt.

We used the GeneChip® *Vitis vinifera* Genome Array which provides comprehensive coverage of the *V. vinifera* (grape) genome to identify significantly differentially expressed (at least two) transcripts ($p$ <0.01) between salt-stressed and control plants or ER-stressed and control plants by microarray analysis. We then screened these transcripts for those meeting or exceeding a twofold expression ratio and obtained a set 2061 differentially expressed transcripts at 6 and 24 h post-treatment for both ER- and salt-stress treatments (S1 Table).

Salt stress had a greater impact than ER stress on the number of transcripts with changed expression. The number of transcripts that differed between ER and salt stresses was significant (Fig 2).

### Analysis of gene expression in response to ER stress

The expression of 456 genes changed in response to ER stress induced by TM (Fig 2A and 2B). The 306 transcripts with increased expression under TM treatment and 150 transcripts with decreased expression under TM treatment are listed in S2 Table. At both 6 and 24 h after TM treatments, the coefficients of the transcripts with increased or decreased expression levels (fold change) were close and most of the common transcripts with decreased expression were found at 24 h ER stress. At 6 h post-stress, probable CCR4-associated factor (1622598_at, 9-fold), unknown (1610721_s_at, 7-fold) and unknown (1611385_at, 7-fold) exhibited the highest increases in expression. At 24 h post-stress, unknown (1607468_at, 15-fold), early nodulin-75-like (1619147_at, 10-fold), B-box zinc finger protein 32 (1610633_at, 10-fold) genes showed the highest increases in expression (S2 Table).

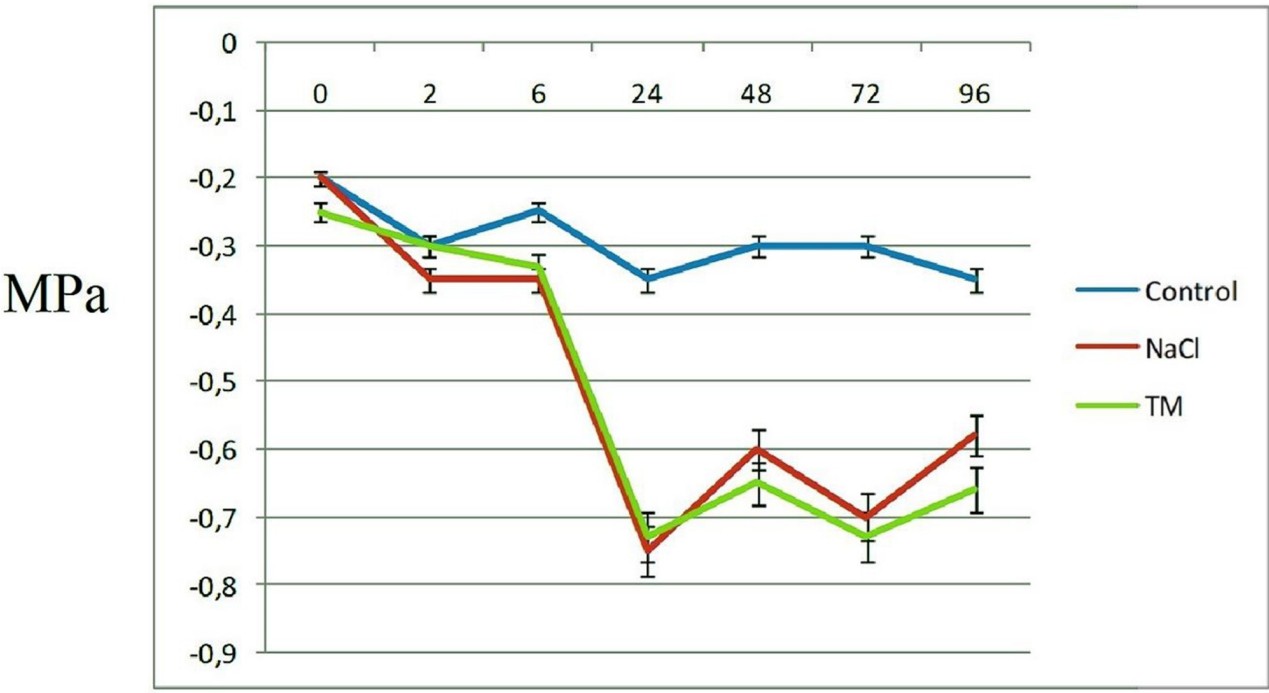

**Fig 1. Stem water potential of leaves treated with salt and ER stresses.** Plantlets were grown in a growth chamber for 10 weeks at 25 °C. Plantlets with 4–5 leaves and 2–4 roots were chosen for use in stress treatments. Measurements were performed for ER and salt stressed samples and controls at indicated time points. Results were statistically analyzed using $t$-tests ($p < 0.05$). Data are means of eight measurements.

We then conducted GO term classification to assign ER stress-responsive transcripts to putative functional categories in the 'biological process' domain (Fig 3A). Transcripts in the 'biological process' (BP) domain at 6 h post-treatment include those that match functional terms 'regulation of transcription' (six genes), 'transmembrane transport' (one gene) (Fig 3A). At 24 h post-treatment, transcripts related to 'regulation of transcription' (ten genes), followed by those related to 'oxidation-reduction process' (twentyone genes), 'transmembrane transport' (thirteen genes), 'metabolic process' (seven genes), and 'protein folding' (four genes) were expressed distinctly (Fig 3A).

In the 'molecular function' (MF) domain at 6 h post-treatment, 6 stress-responsive transcripts match the term 'DNA binding functions' and 4 stress-responsive transcripts match the term 'transcription regulator activity' (Fig 3B). Another two genes in this domain were categorized in the "sequence-specific DNA binding' process' term and two other transcripts were categorized in the 'metal ion binding activity' term. At 24 h post-treatment, nine stress-responsive transcripts were related to 'DNA binding' and sixteen stress-responsive transcripts were related to 'catalytic activity' (Fig 3B). Thus, these pathways might be regulated in response to ER stress (Fig 3B).

In the 'cellular component' (CC) domain, six transcripts related to 'nucleus', three transcripts related to 'integral component of membrane', and another three transcripts related to 'membrane' were identified at 6 h post-treatment. The 'cellular component' terms with the highest numbers of stress-responsive transcripts at 24 h post-treatment were annotated into

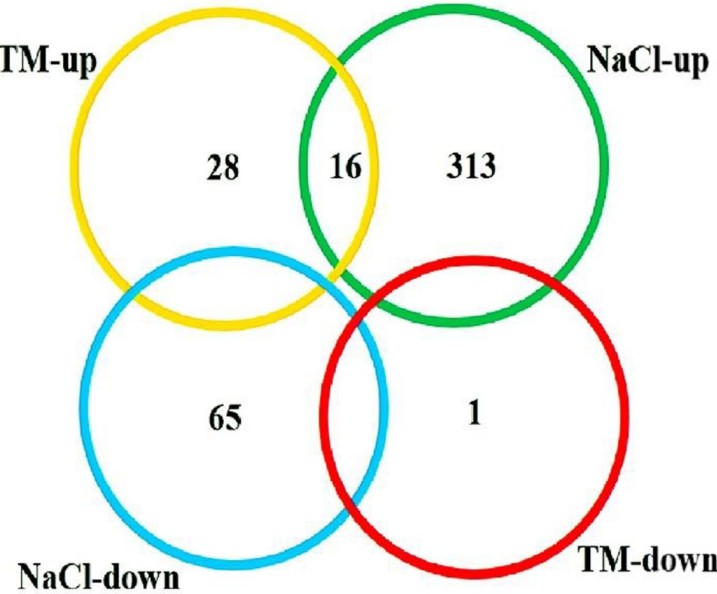

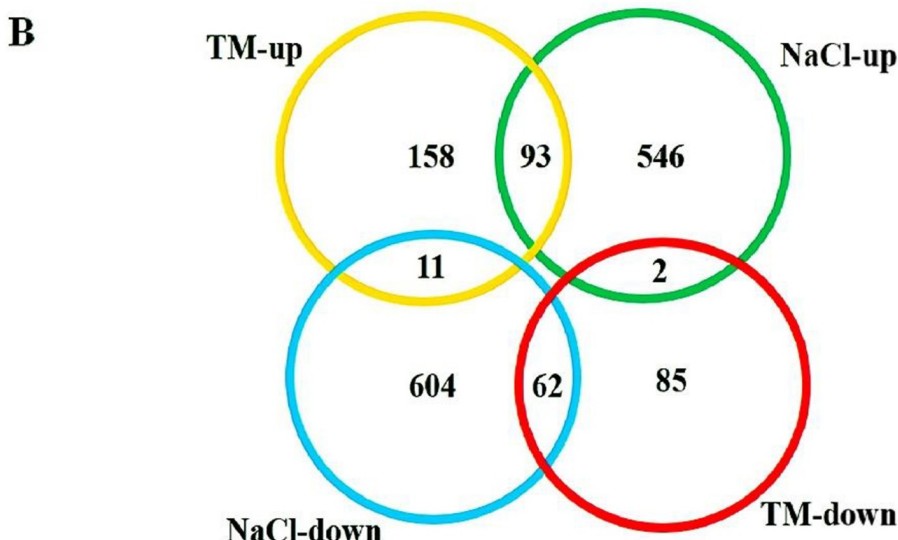

**Fig 2. Venn diagram of microarrays analysis performed on control, NaCl and TM treated plants at 6 h (A) and 24 h (B).** Green circles show transcripts that were at a higher abundance in NaCl treated plants than the control, and blue circles show transcripts that were at a lower abundance in NaCl treated plants than the control. Yellow circles show transcripts that were at a higher abundance in TM treated plants than the control, and red circles show transcripts that were at a lower abundance in TM treated plants than the control. At 6 h post-treatment the abundances of 16 transcripts common to both ER- and salt-stressed conditions differed significantly from their respective controls, while the abundances of 378 transcripts unique to salt stress differed significantly from their control. At 24 h post treatment, 1150 transcripts were unique to salt stress, whereas about 243 transcripts unique to ER stress differed significantly from control.

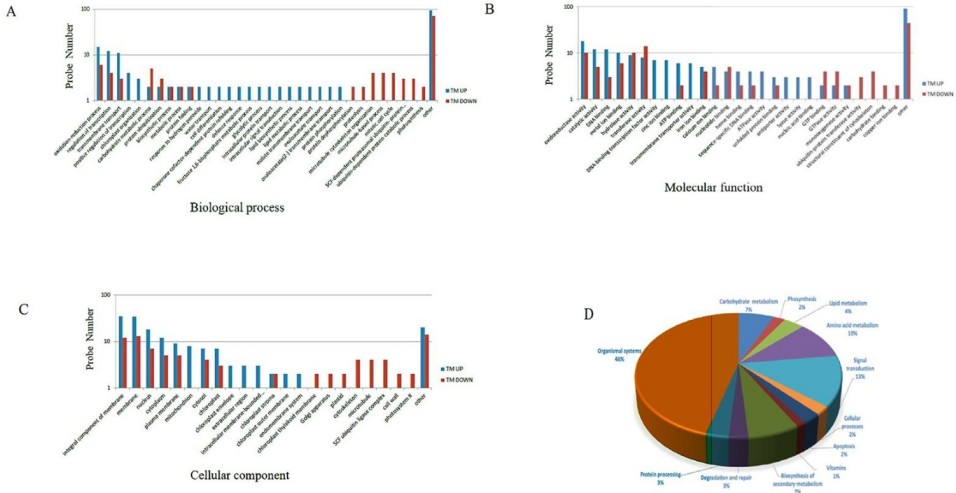

**Fig 3. Gene Ontology and KEGG pathway classification of ER-stress responsive genes.** (**A**) indicates 'biological process' and (**B**) indicates 'molecular functions' (**C**) indicates 'cellular component' for transcripts with increased or decreased expression under ER stress. Each bar represents the number of differentially expressed transcripts in each category (**D**) The most significantly enriched KEGG pathways for ER-stress responsive genes.

'membrane' (44 genes), 'integral component of membrane' (44 genes) and 'chloroplast' terms (10 genes) (Fig 3C).

We also performed KEGG pathway analysis of differentially expressed genes which were divided into different enriched categories, including carbohydrate metabolism (16 genes), amino acid metabolism (24 genes), signal transduction (31 genes), lipid metabolism (9 genes) and organismal systems (108 genes) (Fig 3D; S3 Table).

## Analysis of gene expression in response to salt stress

A total of 742 transcripts significantly decreased and 970 transcripts significantly increased in salt-stressed plants compared to control (S4 Table, Fig 2A and 2B) were classified into functional categories. Many of transcripts with increased or decreased abundance in response to salt stress compared to control encode TFs, which suggests a mechanism for their possible roles in response to salt stress. The abundance of transcripts such as aquaporin TIP1-1 (1607943_at; decreased at least 20-fold) and a unknown gene (1621950_s_at; decreased at least -fold) changed dramatically in salt-stressed rootstocks (S4 Table). Transcripts of other genes such as 9-cis-epoxycarotenoid dioxygenase 1 (1606788_s_at) and two unknown genes (1611272_at and 1615147_at) increased at least 30-, 36-, and 29-fold, respectively, in salt-stressed rootstocks (S3 Table). The transcript expression of unknown gene (1611272_at) increased 36-fold, followed by 9-cis-epoxycarotenoid dioxygenase 1 gene (1606788_s_at), which increased 30-fold, and ubiquinol oxidase 2, mitochondrial gene (1607193_at) which increased 25-fold. Genes encoding germin-like protein (1611611_at, 62 fold), dehydrin (1621592_s_at, 39-fold) and an unknown (1611272_at, 42-fold) showed the highest transcript expression at 24 h post-treatment (S4 Table).

In the biological process domain (BP), most transcripts were categorized into the GO terms 'metabolic process' (24 genes), 'photosynthesis' (31 genes), 'oxidation-reduction process' (90 genes), 'protein phosphorylation' (38 genes), and 'regulation of transcription' (55 genes) (Fig 4A).

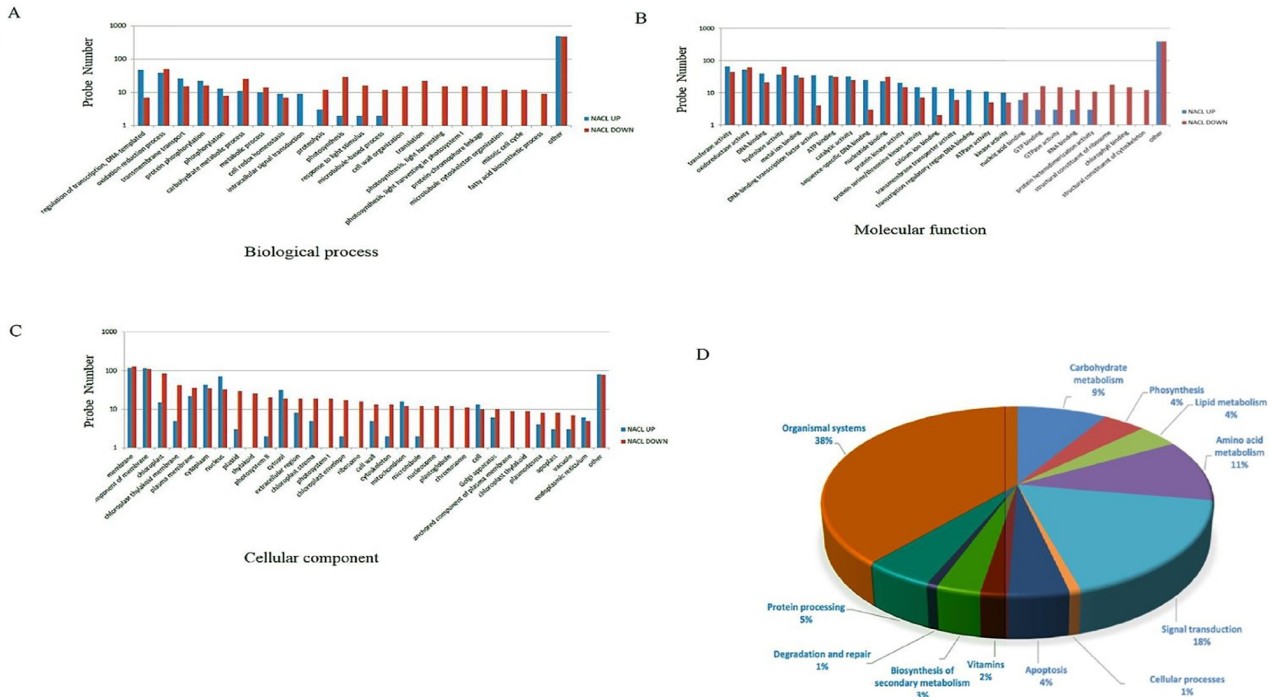

**Fig 4. Gene Ontology and KEGG pathway classification of salt-stress responsive genes.** (**A**) indicates 'biological process' and (**B**) indicates 'molecular functions' (**C**) indicates 'cellular component' for transcripts with increased or decreased expression under salt stress. Each bar represents the number of differentially expressed transcripts in each category (**D**) The most significantly enriched KEGG pathways for salt-stress responsive genes.

In the 'molecular function' (MF) domain, at both 6 and 24 h post-treatments, the most enriched functional categories in salt-treated grapevine were 'catalytic activity' (57 genes), 'hydrolase activity (100 genes), 'transferase activity' (111 genes), 'oxidoreductase activity' (103 genes), 'nucleotide binding' (54 genes), and 'metal ion binding' (64 genes) (Fig 4B).

In the cellular component (CC) domain, at both time points post-treatment, most transcripts were related to 'membrane' (243 genes) followed by 'cellular membrane' (224 genes), while 'nucleus' and 'chloroplast' terms included 104 and 99 transcripts, respectively (Fig 4C).

We also performed KEGG pathway analysis of salt-stress responsive genes which were divided into different enriched categories, including carbohydrate metabolism (76 genes), amino acid metabolism (93 genes), signal transduction (155 genes), lipid metabolism (34 genes), protein processing (43 genes) and organismal systems (334 genes) (Fig 4D; S3 Table).

## Validation of microarray results using qRT-PCR

We performed qRT-PCR analysis of specific transcripts in response to salt and ER stresses to validate our microarray results. We chose to analyze 14 genes from different functional categories including those encoding stilbene synthase, TIFY 10A-like, and ABP19a-like, which were regulated in common by ER and salt stresses according to our microarray data. The expression of 14 selected genes were analyzed by qRT-PCR of RNAs isolated from ER-stressed or salt-treated plants at four different time points with three replicates. As the results in Fig 5 show, the expression of these genes varied at different time points and increased or decreased considerably at 6 and 24 h post-treatments (Fig 5A, 5B, 5C and 5D). Validation of the microarray

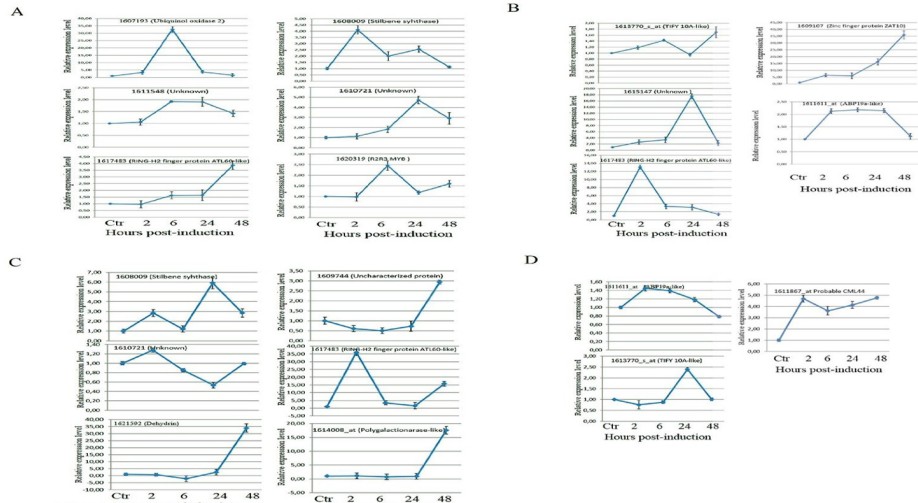

**Fig 5. Expression analysis of chosen genes by qRT-PCR.** Total RNA was isolated from leaves of grapevine rootstock treated with NaCl (**A, B**) or tunicamycin (TM) (**C, D**) for 2. 6. 24. and 48. hours indicated times. The bars represent the relative intensity of labelled qRT-PCR products from three independent biological replicates. *eIF4α* was used as an internal control Genes chosen for analysis of expression by qRT-PCR included 1607193 (ubiquinol oxidase 2), 1608009 (stilbene syhthase), 1611548 (unknown), 1610721 (unknown), 1617483 (RING-H2 finger protein ATL60-like), 1620319 (R2R3 MYB), 1609107 (zinc finger protein ZAT10), 1615147 (Unknown), 1621592 (dehydrin), 1611611_at (ABP19a-like), 1609744 (uncharacterized protein), 1614008_at (polygalactironase-like), 1611867_at (probable CML44), 1613770_s_at (TIFY 10A-like).

data by qPCR for genes whose expression was significantly correlated with microarray data in salt-stressed plants was presented in S1 Fig.

## Comparison of gene expression profiles between salt- and ER-stressed grapevine

We further compared transcript abundances in response to salt and TM treatments. The results showed that 856 transcripts at 6 h post-treatment and 1434 transcripts at 24 h post-treatment changed significantly, at least twofold, in response to salt stress and more than under TM treatment. Among the differentially expressed genes, transcript abundances of 1273 genes increased and those of 1093 genes decreased under both salt- and ER-stress conditions (Fig 2A and 2B).

To determine genes induced in common by both ER and salt stress and possibly related to UPR, we investigated the overlap of the gene sets identified under the two stress treatments. Fig 2A and 2B show the Venn diagram indicating the overlap of gene sets with increased or decreased expression under the two stress treatments. We found that the increased transcript expression of 30 genes and the decreased transcript expression of one gene overlapped between the two stress treatments (Fig 2A and 2B).

These 29 differentially expressed transcripts are predicted to encode proteins such as TFs including WRKY, NAC domain-containing proteins, ethylene-responsive TFs, and ion transporters.

## Differential transcript expression of genes encoding transcription factors in response to salt- and ER-stress treatments

We identified at least 50 TFs that were differentially expressed in response to salt and ER stresses S2 and S4 Tables. These differentially expressed transcripts encode nine WRKY TFs,

NACs, ten ethylene-responsive TFs, eight Ring-H2 finger protein, twenty two zinc finger protein, three heat stress TF, five scarecrow-like protein, an AP2/ERF and two B3 domain. Among these transcripts, especially the expression of the one encoding NAC transcription factor 29 (1621876_at), one B-box zinc finger protein 32 (1610633_at), one zinc finger protein CONSTANS-LIKE 7 (1613070_at) increased more than twofold under both stresses (Table 2).

In response to ER stress, the transcript expression of genes encoding ethylene responsive TFs, MYB-related TFs, and TINY TFs changed differentially (S2 Table). In response to salt stress, the transcript expression of genes encoding zinc finger proteins, NAC domain-containing proteins, homeobox-leucine zipper proteins (HD-Zip), WRKY TFs, and MYB TFs increased in leaves. Among these, the abundances of transcripts encoding R2R3 transcription factor MYB108-like protein 1 and WRKY TFs increased significantly, more than 4-fold and 6-fold, respectively, in response to salt stress (S4 Table).

## Transcript expression of genes involved in carbohydrate metabolism and osmotic adjustment

We identified differentially expressed transcripts involved in carbohydrate metabolism in grapevine rootstock 1616C leaves in response to salt stress. The transcript abundances of four transcripts decreased, while those of genes encoding and UDP-glycosyltransferase (1610410_at and 1618965_at) increased in response to salt stress (S3 Table). Interestingly, the abundance of transcripts encoding xyloglucan endotransglucosylase (1115533_s_at, 1617739_at and 1620096_at) increased in ER-stressed grapevine leaves but decreased in salt-stressed leaves, revealing opposing responses between these two stresses.

Our results revealed a decrease in expression of proline-rich proteins (1606530_s_at; 1615201_at; 1619613_at; 1612791_at; 1607162_s_at) in salt-stressed plants. Additionally, the transcript abundances of amino acid transporters such as the lysine histidine transporter (1617470_s_at) increased significantly in salt-stressed plants (S3 Table). In support of these results, transcript abundances of genes involved in amino acid metabolism such as L-allo-threonine aldolase-like (1610059_at), methionine gamma-lyase-like (1611875_at), amino transferase TAT2-like (1618777_at), serine acetyltransferase 1 (1614014_at), and polyphenol oxidase (1609234_at) increased under salt stress (S3 Table). The transcript expression of glutamine synthetase (1613697_at) decreased under ER stress but did not change under salt stress (Table 2). These results suggest that osmoprotectants such as proline and amino acids are synthesized and transported to maintain cell turgor in response to stress.

## Effects of salt and ER stress on transcript expression of genes involved in photosynthesis, chlorophyll degradation, and glycolysis

In our study, the transcript abundances of genes encoding key enzymes in the Calvin cycle (1620551_s_at) and chlorophyll a-b binding proteins significantly decreased in the salt-stressed rootstocks (S3 Table). Interestingly, the abundance of transcripts of a gene encoding chlorophyll a-b binding protein 13 (1619629_at) decreased in salt-stressed leaves (12-fold) but did not change in ER-stressed plants (Table 2).

The transcript abundances of two genes involved in glycolysis, phosphoglycerate (1608530_at), increased in salt-stressed plants, while that of a gene encoding fructose bisphosphate aldolase genes (1622065_at, 1609678_at, 1616002_s_at, 1621944_at) and malate dehydrogenase (1620184_at) decreased (S3 Table), and interestingly, transcripts of genes encoding fructose bisphosphate aldolase (1621944_at) exhibited opposite expression patterns in response to salt and ER stresses (Table 2).

**Table 2. List of genes regulated in common by both salt and ER stresses.** Fold-variation after 6 and 24 h in salt- and TM- treated or untreated plants for the same gene.

| Affymetrix ID | Gene | TM-UP-6 | TM-UP-24 | TM-DOWN-6 | TM-DOWN-24 | SALT-UP-6 | SALT-UP-24 | SALT-DOWN-6 | SALT-DOWN-24 |
|---|---|---|---|---|---|---|---|---|---|
| **Transcription factors** | | | | | | | | | |
| 1620621_at | NAC domain-containing protein 2 | | 2.18 | | | 2.94 | 2.81 | | |
| 1619573_at | zinc finger protein ZAT10-like | 6.95 | | | | | 2.23 | | |
| 1610633_at | B-box zinc finger protein 32 | | 10.44 | | | | 11.00 | | |
| 1606822_at | zinc finger CCCH domain-containing protein 53 | | | | 2.08 | | | | 2.94 |
| 1613070_at | zinc finger protein CONSTANS-LIKE 7 | | | | 12.65 | | | | 4.34 |
| 1617483_at | RING-H2 finger protein ATL60-like | 4.27 | | | | 5.99 | | | |
| 1606575_at | RING-H2 finger protein ATL74 | | | | 2.94 | | | | 2.85 |
| 1621876_at | NAC transcription factor 29 | 3.62 | 3.08 | | | 2.54 | 4.38 | | |
| 1619311_at | pathogenesis-related genes transcriptional activator PTI5-like | 2.42 | | | | 3.62 | 2.77 | | |
| **Glucose metabolism** | | | | | | | | | |
| 1621944_at | fructose-bisphosphate aldolase 2, chloroplastic-like | | 3.66 | | | | | 2.85 | |
| 1609678_at | fructose-bisphosphate aldolase, cytoplasmic | | | | | | | 5 | |
| 1620628_at | neutral invertase | | | | 2.75 | | | | |
| **Unknown** | | | | | | | | | |
| 1610210_at | uncharacterized LOC100263114 | | | | | | | 3.5 | |
| 1608050_at | uncharacterized LOC100242324 | | 3.21 | | 4.45 | 3.99 | | | |
| 1608132_at | uncharacterized LOC100265554 | | 2.42 | | | | 3.22 | 5.55 | |
| 1614931_at | uncharacterized LOC100241227 | | 2.9 | | 2.53 | 2.49 | | | |
| 1615141_at | uncharacterized LOC100247840 | | 4.19 | | | | | 3.50 | |
| 1618889_at | uncharacterized LOC100266582 | | | 3.33 | | | | 5.00 | |
| 1620072_s_at | uncharacterized LOC100261167 | | 4.09 | | | 2.18 | | | |
| 1622785_at | uncharacterized LOC100255423 | 3.62 | 3.08 | | 2.54 | 4.38 | | | |
| 1615141_at | uncharacterized LOC100247840 | | 4.19 | | | | | 3.50 | |
| 1611705_s_at | uncharacterized | | | 6.66 | | | | 2.86 | |
| 1607155_at | uncharacterized LOC100266884 | | 2.17 | | | | | 2.04 | |
| **Signal transduction** | | | | | | | | | |
| 1611867_at | probable calcium-binding protein CML44-like | | 7.18 | | 2.56 | 5.25 | | | |

(*Continued*)

**Table 2.** (Continued)

| Affymetrix ID | Gene | TM-UP-6 | TM-UP-24 | TM-DOWN-6 | TM-DOWN-24 | SALT-UP-6 | SALT-UP-24 | SALT-DOWN-6 | SALT-DOWN-24 |
|---|---|---|---|---|---|---|---|---|---|
| 1612124_at | caffeic acid O-methyltransferase | | 3.43 | | 9.86 | | | | |
| 1621563_x_at | caffeic acid O-methyltransferase | 3.35 | | | 9.03 | 7.87 | | | |
| **Transporters** | | | | | | | | | |
| 1613603_at | lysine histidine transporter-like 8-like | 2.64 | | | 3.21 | | | | |
| 1610696_s_at | probable transporter mch1 | | 3.20 | | 6.60 | | | | |
| 1608991_at | probable plastidic glucose transporter 2 | | 2.41 | | 3.60 | 3.27 | | | |
| 1610527_at | polyol transporter 5 | | 2.38 | | | 2.20 | | | |
| **Hormone regulation** | | | | | | | | | |
| 1611569_at | chitin-inducible gibberellin-responsive protein 1-like | | 2.18 | | 3.40 | 2.32 | | | |
| 1610607_at | gibberellin-regulated protein 4 | | | 5.26 | | | | 8.33 | |
| **Stres response** | | | | | | | | | |
| 1615001_s_at | salt stress-induced hydrophobic peptide ESI3 | | 5.15 | | 2.50 | 2.33 | | | |
| 1621592_s_at | dehydrin | | 2.88 | | 9.57 | 39.12 | | | |
| 1616865_a_at | universal stress protein A-like protein | 2.80 | | | | 2.96 | | | |
| 1612386_at | cytochrome P450 82C4-like | | 2.28 | | | | 2.03 | 3.81 | |
| 1610325_at | cytochrome P450 71A26-like | | 3.35 | | | 5.30 | | | |
| 1613695_at | cytochrome P450 71A26-like | | 2.02 | | | 2.51 | | | |
| 1612325_at | cytochrome P450 86B1 | | | 3.33 | | | | 8.33 | |
| 1615246_at | cytochrome P450 86A7 | | | 4.34 | | | | 7.69 | |
| 1619565_at | pyrroline-5-carboxylate synthetase | | 2.57 | | | 2.60 | | | |
| **Cell redox homeostasis** | | | | | | | | | |
| 1619667_at | DNA-damage-repair/toleration protein DRT100-like | | | 6.25 | | | | 7.69 | |
| **Glycine metabolism** | | | | | | | | | |
| 1607541_at | glycine-rich cell wall structural protein 2-like | | 3.80 | | 2.88 | 3.88 | | | |
| 1609989_at | glycine-rich cell wall structural protein 1-like | | | 4.00 | | | | 2.85 | |
| **Vesicle fusion** | | | | | | | | | |
| 1608365_at | syntaxin-121-like | | 2.52 | | | 2.18 | | | |
| **Photosynthesis** | | | | | | | | | |
| 1622208_x_at | probable oxygen-evolving enhancer protein 2 | | | 5.55 | | | 3.33 | | |
| 1610646_s_at | beta-amylase 1, chloroplastic | | 6.02 | | 6.15 | 6.77 | | | |
| 1606705_at | chaperone protein dnaJ 11, chloroplastic | | 3.09 | | 4.22 | 3.44 | | | |
| 1622622_a_at | carboxyl-terminal-processing peptidase 3, chloroplastic | | 2.94 | | | 3.55 | | | |

(*Continued*)

**Table 2.** (Continued)

| Affymetrix ID | Gene | TM-UP-6 | TM-UP-24 | TM-DOWN-6 | TM-DOWN-24 | SALT-UP-6 | SALT-UP-24 | SALT-DOWN-6 | SALT-DOWN-24 |
|---|---|---|---|---|---|---|---|---|---|
| 1609491_at | carboxyl-terminal-processing peptidase 3, chloroplastic | | 2.69 | | | 3.47 | | | |
| 1619329_at | chloroplastic group IIB intron splicing facilitator CRS2-B, chloroplastic | | 2.50 | | | | | 2.04 | |
| 1614037_at | chaperonin 60 subunit beta 4, chloroplastic | | | 3.33 | | | | 3.57 | |
| 1612311_at | phosphomethylpyrimidine synthase, chloroplastic | | | 4.54 | | | | 2.70 | |
| 1614175_at | thiamine thiazole synthase 2, chloroplastic | | | 5.26 | | | | 4.00 | |

## Effects of salt stress and ER stress on transcript expression of genes encoding transporters

The expression profiles of several genes encoding various transporters were significantly affected by both ER and salt stresses compared to controls. A total of 42 genes that were more than twofold differentially expressed during salt stress were classified as transporters (S3 Table). Among these, 27 genes exhibited increased expression and 15 genes exhibited decreased expression during salt stress in grapevine. In ER-stressed plants, we identified two genes with differentially increased expression unique to ER stress and four genes with differentially altered expression under both salt and ER stresses (Table 2). For example, we found that the abundances of transcript encoding lysine histidine transporter-like-8 (1613603_at) all increased under both salt and ER stresses (Table 2).

## Differences in transcript expression of stress-responsive genes

The transcript expression of seven stress-responsive genes including those encoding galactinol synthase (1615552_at), glutaredoxin-C9-like (1616703_at), syntaxin-24-like (16116643_at), and tropinone reductase (1614862_at) increased in salt-stressed leaves (S3 Table). In contrast, a total of three stress-responsive genes including MLP-like protein 28-like (1616133_at), and stress responsive A/B barrel domain (1618245_at) decreased in response to salt stress. Interestingly, two stress-responsive transcripts including universal stress protein A-like protein (1616865_a_at) and dehydrin (1621592_s_at) were differentially expressed in response to ER and salt stresses (Table 2).

## Differential expression of genes encoding plant hormone-related functions

We identified more than nine differentially expressed genes encoding hormone-related functions in response to salt stress, and three such genes in response to TM-induced ER-stress. Additionally, four genes encoding hormone-related functions were regulated in common by both salt and ER stresses. Among these genes, the transcript expression of two genes encoding 9-*cis*-epoxy carotenoid dioxygenase (NCEDs; 1606788_s_at; 1608022_at) and one gene encoding β-carotene hydrolase (1619371_at) increased significantly, more than 41- and 30-fold, and 18-fold, respectively, in response to salt stress (S2 Table). In TM-treated plants, the transcript expression of one gene encoding ABA metabolism-related functions increased at least two fold compared to control plants (S2 Table). In the present study, the transcript expression of three ethylene-responsive TFs (1619390_at; 1617671_s_at; 1616198_at and 1606975_at) significantly

increased in response to ER stress. However, in salt-stressed rootstocks, the transcript abundance of at least three ethylene biosynthesis-related genes (1608961_at; 1609629_at; 1612636_at) increased (S2 and S4 Tables).

In addition, we identified six auxin-responsive genes expression in response to salt stress. Among these, the transcript abundance of one gene encoding auxin-binding protein auxin-induced protein X10A (1609828_at) increased while that of auxin-binding protein ABP19a precursor (1613054_at; 1606566_at), auxin-responsive protein IAA16-like (1619741_at; 1621754_at) and auxin-induced protein 22D (1611491_at) decreased more than 2-fold under salt treatment (S4 Table). The transcript expression of one gene encoding gibberellin-regulated protein 4-like (1610607_at) significantly decreased at least eightfold in response to salt stress (S4 Table).

## Differential expression of genes related to ER stress

Biotic and abiotic stresses including salt or heat stress and treatments with agents such as TM can induce ER stress [45]. In support of previous findings, we found that five transcripts predicted to encode ER stress-related genes were differentially expressed in salt-treated leaves. These transcripts include those that encode PDI (protein disulfide isomerase) (1616531_at), mitochondrial protein (1616729_at), heat shock factor protein HSF24 (1616889_at), HSF24-like (1612310_at) (Table 2). Interestingly, the transcript abundances of biotic and abiotic stress-regulated genes such as WRKY transcription factor 65-like (1606659_s_at), stilbene synthase 1-like (1620964_s_at), RING-H2 finger protein (1617483_at), and WRKY transcription factor 40-like (1614806_s_at) also increased in response to ER stress (S2 Table, Table 2).

## Effects of salt and ER stress on the expression of genes involved in cell wall modification

In the present study, the transcript expression of a number of genes related to cell wall composition significantly decreased in response to salt stress or ER stress including, pherophorin-C5 protein precursor, pistil-specific extensin-like protein, and expansin-like proteins. In addition, the transcript expression of genes encoding esterase-like protein and expansin B1 protein significantly increased only in salt-stressed rootstocks (S4 Table).

The composition of fatty acids changes in plants under stress conditions [46, 47]. In the present study, the expression of transcripts related to fatty acid metabolism increased strongly in response to salt stress (S4 Table). These include genes encoding malonate—CoA ligase (1609153_at), and two genes encoding 12-oxophytodienoate reductase 11 (1621543_x_at).

## Effects of salt and ER stress on the expression of genes involved in chromatin structure

In the present study, the transcript abundances of 13 genes encoding histone proteins decreased significantly under salt stress in the grapevine rootstock 1616C (S3 Table). Among these, the expression of histone H2AX (1608056_at), histone H2A (1608927_at), histone H4 (1610096_at; 1613076_at), histone H3.2 (1612573_at; 1622440_at; 1614219_at; 1620332_at; 1612573_at; 1622440_at) decreased more than threefold compared to controls.

## Expression of genes encoding putative phosphatase and protein kinase

We found 33 transcripts encoding protein kinases and 16 encoding phosphatases that were differentially expressed in response to salt stress (S3 Table). Among these, expression of transcripts encoding cytokinin riboside 5'-monophosphate phosphoribohydrolase LOG5-like

(1608039_s_at) and phosphatase 2C 24-like (1608253_at) protein increased more than eight-fold under salt stress, while expression of transcripts encoding a serine/threonine protein kinase SAPK3-like (1618638_at) and a CBL-interacting protein kinase (1611172_at) increased more than fourfold under salt stress (S3 Table).

## Discussion

In the present study, we identified genes encoding certain TFs and those in pathways such as hormone synthesis, ion transport, lipid metabolism, carbohydrate metabolism, photosynthesis, and ER stress metabolism involved in salt- and ER-stress responses in grapevine rootstock 1616C. These results provide new insights into the molecular mechanisms of the relationship between salt- and ER-stress responses in plants.

### Salt stress treatment also induces expression of ER stress-responsive genes

There is some indirect evidence indicating that ERAD occurs in plants [48, 49]. Recently, Liu et al. [28] have demonstrated that salt stress induces ubiquitination of proteins and that the ERAD pathway is necessary for plants to tolerate salt stress. Consistent with these results, we found that the transcript expression of E3 ubiquitin-protein ligase (1607728_at), increased at least threefold under both salt- and ER-stress treatments.

Molecular chaperones and foldases are a class of proteins that bind to misfolded or unfolded proteins to facilitate their proper folding under stress conditions [50]. When protein disulfide isomerase (PDI) which catalyzes disulfide bond formation, is absent, pro-glutein accumulation is enhanced in the ER and ER stress is induced [50]. Heat-shock proteins (HSPs) are important molecular chaperones. They regulate protein folding, assembly, translocation, and degradation under stress conditions [51, 52]. Here, salt-stress induced expression of transcripts encoding HSF 24-like (1612310_at), and class I heat shock proteins (1616145_a_at; 1609554_at; 1613042_at), which suggests that these proteins might be involved in refolding of misfolded proteins to relieve ER stress.

### Transcription factors involved in ER and salt stresses

Several TFs such as AP2/ERF, bZIP, NACs, WRKY, MYB, and HD-Zip (ATHB) proteins can control the expression of target genes by binding to specific *cis*-acting elements in the promoters of these genes [53], and some TFs can confer resistance to drought or salt stress in plants [54, 55]. Interestingly, In the present study, the expression of transcripts encoding TFs such as WRKY, MYB, NAC and bZIP increased significantly in response to salt stress, which suggests that they might regulate the synthesis and accumulation of proline, LEA proteins, and sugars that are key metabolic markers of stress tolerance [56].

Similarly, the transcript expression of WRKY transcription factors increased at least two-fold under both ER and salt stresses. These findings support a relationship between salt stress and ER stress, and suggest that these TFs might play key roles as positive regulators of plant tolerance of abiotic stress.

The NAC TFs are involved in regulation of responses to different stress factors such as cold, drought, high salinity, osmotic stress, or hormonal factors [23, 57, 58]. For instance, overexpression of NAC TFs in rice and *Arabidopsis* confers tolerance to salt stress [54]. In addition, NAC17, NAC062, NAC089, and NAC103 are activated in response to ER-stress inducers such as TM (tunicamycin) and DTT (dithiothreitol) [23, 57]. In accordance with these previous results, the expression of NAC 21/22-like (1610480-like) and NAC 72-like (1609172_at) transcripts increased in salt-stressed grapevine. In addition, expression of transcripts encoding the NAC transcription factor 29 (1621876_at) and NAC domain-containing protein 2

(1620621_at) increased under both ER and salt stresses. These results suggest that a number of NAC domain-containing genes might be involved in salt- and ER-stress responses.

Liu et al. [59] identified a signaling pathway in plants that responds to salt stress. This pathway includes an ER membrane-bound TF, AtbZIP17, which is cleaved by AtS1p and then translocated into the nucleus where it activates stress-response genes [59]. In our study, we observed an increased abundance of transcripts encoding a bZIP protein (1619664_at) under salt stress.

The expression of a number of zinc finger proteins such as ZFP179, ZFP182, and ZFP252 is reportedly induced by salt stress [60]. In the present study, we found that the abundance of transcripts encoding two zinc finger proteins (1619573_at and 1610633_at) increased under both ER and salt stresses, while the abundance of transcripts encoding a zinc finger A20 (1613186_at and 1615596_s_at), a zinc finger ZAT-11- like (1614332_s_at), and a zinc finger ZAT-10- like (1619573_at) increased only under salt stress. These findings suggest that certain zinc finger proteins might be involved in both ER and salt stress responses.

MYB TFs play important roles in controlling cellular processes and are involved in responses to biotic and abiotic stresses, development, and disease defense [61]. Here, the expression of an R2R3Myb108 TF (1607133_at), a MYB44-like (1622782_at) and another MYB59 (1609021_at) was strongly increased by salt stress. The above findings suggest that genes encoding MYB TFs in the *Vitis* genome and R2R3-MYB family proteins might be involved salt-stress responses in grapevine.

RING (Really Interesting New Gene) zinc finger proteins have crucial roles in regulating plant development and plant-environment interactions [62]. The RING finger motif mediates protein-protein interactions and is essential for E3 ubiquitin ligase activity [63]. Interestingly, a RING-H2 finger protein (1617483_at) was strongly induced in response to both ER and salt stresses, suggesting a linkage between ER and salt stresses in plants. If salt stress does induce ER stress, this TF might be involved in protein ubiquitination in response to ER stress.

Recent studies reported that several ERFs (Ethylene Response Factors) bind to dehydration-responsive promoter elements and are key regulators of abiotic stress tolerance [64]. For instance, AtERF1 and AtERF4 were induced by salt stress [65, 66]. Similarly, CaERFP1 and BrERF4 confer salt tolerance in tobacco and *Arabidopsis* [65, 67]. Interestingly, we found that the abundance of transcripts encoding ERF4-like transcription factor (1606975_at) increased in response to ER-stress treatments, indicating a possible association between these two stresses.

In plants, histone modification and DNA methylation are correlated with changes in the expression of stress-responsive genes to mediate plant adaptation to environmental stresses [68]. For example, salt stress activates HATs (define) and increases global histone acetylation levels in maize genome [68]. In the present study, the transcript abundances of 13 genes encoding histone proteins decreased significantly in grapevine rootstock 1616C under salt stress, which might indicate roles in regulating gene expression in response to salt stress.

## Effects of salt and ER stress on the expression of genes involved in osmotic regulation

Upon exposure to salt stress, plants accumulate compatible solutes such as proline [53, 69], glycine betaine [70], sugars, and polyols [71, 72]. In our study, we observed decreased transcript expression of genes encoding a proline-rich cell wall protein in response to salt stress. Concentrations of amino acids such as cysteine, arginine, and methionine decrease in response to salt stress, while the concentration of proline increases when plants are exposed to salt stress [73]. In the present study, we observed increased abundances in response to salt stress of transcripts

encoded by many genes involved in cysteine, glutamate, aspartate, and tyrosine biosynthesis and in glutamate degradation. Interestingly, transcripts of gene encoding caffeic acid O-methyltransferase (1621563_x_at and 1612124_at), which are involved in polyamine metabolism, were differentially expressed in response to both ER and salt stresses. Therefore, we could hypothesize that these osmolyte solutes might contribute to osmotic adjustment of cells and that activation of amino acid metabolism/synthesis might function in salt stress resistance. Additionally, we observed increased abundance of transcripts encoding amino acid transporters such as the lysine-histidine transporter (1613603_at) under salt stress.

$Na^+/K^+$ homeostasis plays important roles in plant salinity stress tolerance [74]. In *Arabidopsis*, NADPH oxidases such as AtrbohD and AtrbohF also regulate $Na^+/K^+$ homeostasis under salt stress [74]. Similarly, we found that the abundance of transcripts from genes encoding NADPH oxidoreductase (1618893_at) increased in response to salt stress. Here, transcriptome analysis revealed that the expression of transcripts encoding $K^+$ transporters, peptide/nitrate transporters, and ABC transporter C family member 10-like (1617849_at) increased under salt stress. In addition, the abundance of transcripts encoding an inorganic phosphate transporter 2–1 (1620639_at) decreased in response to salt stresses. Genes encoding dehydrin and aquaporin have also been related to salt stress [75]. Consistent with these previous findings, we found that the transcripts of at least five genes encoding aquaporins (1607943_at; 1615829_s_at; 1611312_s_at; 1612244_s_at; 1617599_at; 1610603_at) decreased in abundance in response to salt stress.

## Hormonal regulation of ER and salt stresses

ABA and auxin play important roles in plant responses to environmental stresses such as drought and salinity [76, 77]. ABA regulates xylem water potential as well as water uptake in salt stressed plants [78]. Here, the transcript abundance of several genes strongly increased in response to ER (S2 Table) and salt stresses (S3 Table) indicating that ABA might regulate grapevine responses to both stresses.

The UPR also plays an important role in processes mediated by plant hormones, which might explain the influence of the UPR on plant growth and development. Chen et al. [79] recently found that ER-stress downregulates the expression of genes encoding ER- and PM-localized auxin efflux transporters (PIN-formed or PIN proteins) and the auxin receptors TIR1/AFB. Consistent with this observation, the transcript expression of an auxin efflux carrier component 1-like was downregulated in response to ER stress, indicating some cross talk between auxin and ER stress. In addition, our data indicated that transcript expression of an auxin-responsive protein IAA16-like, auxin-responsive protein IAA27-like was regulated by salt stress.

Changes in concentrations of ethylene and its precursor ACC have been induced by salinity and other stresses [80]. In cotton, several ACOs, another ethylene biosynthesis enzyme, were upregulated under salt treatment [81]. Here, we identified increases in the transcript abundance of three genes encoding ACO (1609995_s_at; 1616698_at; 1622147_at) in response to salt-stress treatment. Thus, we can hypothesize that ethylene concentrations in grapevine could be influenced via ACOs under high-salinity conditions. The expression of ethylene signaling genes is regulated by salinity and other stresses [82]. In cotton, the expression of transcripts encoding several ethylene receptors (ETR1, ETR2, and EIN4), ethylene signaling genes (CTR1, EIN3, ERF1, and ERF2), and MAPK cascade genes (MEKK1-MKK2-MPK4/6) all increased under salt treatments [81]. For instance, overexpression of ERF1 (ethylene-responsive element binding factors) enhanced plant (*Triticum aestivum, Arabidopsis thaliana*) tolerance to salt, drought, and heat stress [66]. We found that transcript expression of ethylene-

responsive TF4-like (1616198_at; 1606975_at; 1617671_s_at) and another ethylene-responsive TF (1608961_at; 1609629_at; 1612636_at) increased under both ER and salt stress, suggesting a possible interaction between ethylene signaling and both of these stresses.

## Effects of salt and ER stress on the expression of genes involved in carbon metabolism

Salinity stress is known to diminish photosynthesis and intensify phosphorylation in grapevine [83]. Plants regulate carbon metabolic pathways such as glycolysis, photosynthesis, and the TCA cycle differently during salt stress. In grapevine rootstock 1616C, transcripts encoding one UDP-glycosyltransferase (1610410_at), two glycosyltransferase (1618965_at;1610410_at), six fructose bisphosphate (1615874_at; 1614023_at; 1622065_at; 1609678_at; 1621944_at; 1616002_s_at) four xyloglucan endotransglucosylase (1615533_s_at; 1617739_at; 1621251_s_at; 1620096_at) and one galactosyltransferase 6-like (1616116_at) were differentially expressed under salt stress. Further, transcripts encoding fructose-bisphosphate aldolase (1621944_at; 1616002_s_at), was differentially expressed under both ER and salt stresses.

RuBisCO activase, another important enzyme, regulates photosynthesis [84, 85] in response to environmental stress. As photosynthesis decreases under salt stress, decreased expression of transcripts encoding RuBisCO activase (1614740_at; 1606590_at; 1620551_s_at) might be attributable to reduced photosynthetic rates in salt stressed plants. Consistent with these results, we also observed decreased expression of fifteen chlorophyll a-b binding protein 13 and oxygen evolving enhancer protein 2 (1607516_at;1613494_s_at; 1609656_s_at) in salt stressed plants.

## Effects of salt and ER stress on the expression of genes involved in lipid metabolism and ROS detoxification

Fatty acid metabolism might play roles in plant adaptation to drought stress if increased levels of certain fatty acid maintain the fluidity and stability of cellular membranes during stress [47, 86]. For example, Zhong et al. [86] showed that lipid degradation processes are activated and total lipid quantities decrease in response to salt stress. Consistent with these finding we observed decreased expression of transcripts encoding proteins (1606656_at; 1614230_at; 1613022_s_at; 1622416_at 1617745_at) that might be involved in lipid transport processes in salt-stressed plants. Additionally, the transcript expression of genes involved in the acetyl CoA pathway (1616853_at; 1614240_at) increased under both ER and salt stress. During lipid degradation, these enzymes could mediate conversion of the resulting acetyl CoA.

The expression of glutathione *S*-transferases (GTSs) in plants can be induced by diverse environmental stresses. GTSs function to maintain cell redox homeostasis and protect plants from oxidative stress [87]. Tobacco seedlings overexpressing a GST-encoding gene, *GaGST* from *Glycine soja*, exhibited enhanced salt tolerance [88]. In the present study, transcript abundance of one GST (1613204_at) increased in response to salt stress. We also observed that transcripts of genes encoding other proteins involved in ROS detoxification, such as phospholipid hydroperoxide glutathione peroxidase, also increased under salt stress.

## Conclusions

Even though the ER is a key organelle for stress-adaptive responses in plants, the transcript expression of genes involved in the ER- and salt-stress responses in grapevine had not been investigated and compared to date.

Here, we present a comparative transcriptome analysis of response to ER and salt stresses in a salt-tolerant grapevine rootstock. The expression patterns of chosen genes with changed expression patterns were then confirmed by qRT-PCR.

Gene Ontology analysis of many genes with stress-induced changes in expression suggested putative roles in the functional categories 'oxidative stress', 'protein folding', 'transmembrane transport', 'protein phosphorylation', 'lipid transport', 'proteolysis', 'photosynthesis', and 'regulation of transcription'. Our transcriptome profiling analysis revealed that many of these genes were regulated in common under both ER and salt stresses. Finally, comparison of the transcriptome profiles of plants exposed to salt or ER stresses suggests that salt stress might be able to induce ER stress. To our knowledge, this study is the first to provide new insights into the abiotic stress response mechanisms that might underlie high salt tolerance in a grapevine rootstock.

## Supporting information

**S1 Fig. Validation graphics of microarray data by Quantitative Real-Time PCR (qRT-PCR) for salt stress at 6h (A) and 24h (B).**
(JPG)

**S1 Table. List of genes differentially regulated by both salt and ER stresses.** Fold-variation after 6 and 24 h in TM- or salt-treated and untreated plants for the same gene.
(XLS)

**S2 Table. List of genes with increased or decreased expression under ER stresses.** Fold-variation after 6 and 24 h in TM-treated or untreated plants for the same gene.
(XLS)

**S3 Table. KEGG pathway analysis data for salt and ER stresses.**
(XLS)

**S4 Table. List of genes upregulated and downregulated by salt stresses.** Fold-variation after 6 and 24 h in salt treated or untreated plants for the same gene.
(XLSX)

## Author Contributions

**Conceptualization:** Birsen Çakır Aydemir, Ali Ergül.

**Data curation:** Birsen Çakır Aydemir, Umut Kibar, Ali Ergül.

**Formal analysis:** Birsen Çakır Aydemir, Umut Kibar, Ali Ergül.

**Funding acquisition:** Birsen Çakır Aydemir, Ali Ergül.

**Investigation:** Birsen Çakır Aydemir, Canan Yüksel Özmen, Umut Kibar, Filiz Mutaf, Pelin Burcu Büyük, Melike Bakır, Ali Ergül.

**Methodology:** Birsen Çakır Aydemir, Canan Yüksel Özmen, Umut Kibar, Filiz Mutaf, Pelin Burcu Büyük, Melike Bakır, Ali Ergül.

**Project administration:** Birsen Çakır Aydemir, Ali Ergül.

**Resources:** Birsen Çakır Aydemir, Ali Ergül.

**Software:** Umut Kibar, Ali Ergül.

**Supervision:** Ali Ergül.

**Validation:** Birsen Çakır Aydemir.

**Visualization:** Birsen Çakır Aydemir, Umut Kibar, Ali Ergül.

**Writing – original draft:** Birsen Çakır Aydemir.

**Writing – review & editing:** Ali Ergül.

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
