## [Decision Letter · Decision Letter 0]

9 Mar 2020

PONE-D-20-03767

Salt stress induces endoplasmic reticulum stress-responsive genes in a grapevine rootstock

PLOS ONE

Dear Ali ERGÜL,

Thank you for submitting your manuscript to PLOS ONE. After careful consideration, we feel that it has merit but does not fully meet PLOS ONE’s publication criteria as it currently stands. Therefore, we invite you to submit a revised version of the manuscript that addresses the points raised during the review process.

We would appreciate receiving your revised manuscript by 28 March 2020. To enhance the reproducibility of your results, we recommend that if applicable you deposit your laboratory protocols in protocols.io, where a protocol can be assigned its own identifier (DOI) such that it can be cited independently in the future. For instructions see: http://journals.plos.org/plosone/s/submission-guidelines#loc-laboratory-protocols

We look forward to receiving your revised manuscript.

Kind regards,

Mayank Gururani

Academic Editor

PLOS ONE

Journal Requirements:

2. We note that you are reporting an analysis of a microarray, next-generation sequencing, or deep sequencing data set. PLOS requires that authors comply with field-specific standards for preparation, recording, and deposition of data in repositories appropriate to their field. Please upload these data to a stable, public repository (such as ArrayExpress, Gene Expression Omnibus (GEO), DNA Data Bank of Japan (DDBJ), NCBI GenBank, NCBI Sequence Read Archive, or EMBL Nucleotide Sequence Database (ENA)). In your revised cover letter, please provide the relevant accession numbers that may be used to access these data. For a full list of recommended repositories, see http://journals.plos.org/plosone/s/data-availability#loc-omics or http://journals.plos.org/plosone/s/data-availability#loc-sequencing.

3. Please include your tables as part of your main manuscript and remove the individual files. Please note that supplementary tables (should remain/ be uploaded) as separate "supporting information" files

Reviewers' comments:

Reviewer's Responses to Questions

**Comments to the Author**

1. Is the manuscript technically sound, and do the data support the conclusions?

Reviewer #1: Yes

Reviewer #2: Yes

Reviewer #3: Yes

2. Has the statistical analysis been performed appropriately and rigorously? 

Reviewer #1: I Don't Know

Reviewer #2: N/A

Reviewer #3: Yes

3. Have the authors made all data underlying the findings in their manuscript fully available?

Reviewer #1: Yes

Reviewer #2: Yes

Reviewer #3: No

4. Is the manuscript presented in an intelligible fashion and written in standard English?

Reviewer #1: Yes

Reviewer #2: No

Reviewer #3: No

5. Review Comments to the Author

Reviewer #1: The authors present their work comparing the transcriptomes of salt and Tm treated grapevine. This data-set has the potential to inform future work and may prove a valuable contribution to the field. On the whole, the author’s discussion of their results is well measured and appropriate for the level of evidence they provide. I do however feel that the manuscript could be improved if the authors were to provide additional information on the experimental set-up. In addition, communication of their results could be improved through changes in the way they present their data. I have provided further specific details in the attached word document

Reviewer #2: Dear Authors,

the experiment is interesting and therefore the article should be improved technically by shortening of reference list at least by 30% (in Reviewer opinion the list is too long) as well as the text. The text needs some reorganizations as well. At present, the Results contains some parts of Methods and Discussion contains some part of Results. Also for the Readers, it would be easier to follow the text when it is connecting reasons with effects by sentence construction using words i.e: because, therefore, for the reason, as a result. At present, the text is a set of separate paragraphs. And paragraphs, especially in Discussion are lists of examples without or with weak explanations/discussion. So please make the text shorter, clearly organized into sections and fluent.

Reviewer #3: The authors compared the effect of salt and tunicamycin-induced ER stress on transcriptome pattern in grapevine rootstock. The experimental system is interesting and appropriate to determine the regulatory mechanism specific either to salt or ER stress or those ones which are involved in the response to both stresses. However, the manuscript is not well-written since a large part is descriptive and the various paragraphs are not connected to each other. It is also too long, and it could be reduced by 30%. It would be interesting to check which target genes of the stress-induced transcription factors were also affected. Based on this analysis, a model of regulatory network with specific and overlapping parts could be created. An enrichment analysis of the expression data could show which regulatory and metabolic pathways were mainly affected by salt and ER-stress or both of them. The abstract should not only list the genes, but try to explaine some regulatory mechanisms. There are some very short paragraphs (1-2 sentences) in the introduction. Longer ones should be created which allow also their better connection. Only the most interesting results should be described in the text and the greatest differences and similarities between the two types of stress should be emphasized. In the discussion the possible regulatory mechanism should be explained based on the present observations at transcriptome level and those ones available in the literature at proteome or metabolome level. Its certain parts are repetition of the results.

Remarks:

1. Abstract – salt stress influences ER stress based on genes induced by both of them. The sentence should be rewritten more carefully since this statement was not proved.

2. Why 5 µg/ml tunicamycine concentration was chosen? It is possible that it was too low since only few genes were affected compared to salt stress.

3. The length of treatments should be described in the method part.

4. The availability of the microarray data is not given.

5. In the second paragraph of the results two numbers for differentially expressed genes are mentioned but the selection criteria is not given for the first one.

6. The correlation between microarray and qRT-PCR data should be calculated.

7. The activation of Pro synthesis is mentioned during the investigated stresses, but the related genes (P5CR, P5CS) are not listed among the affected ones.

8. The typing and grammatical errors should be corrected.

6. PLOS authors have the option to publish the peer review history of their article (what does this mean?). If published, this will include your full peer review and any attached files.

Reviewer #1: Yes: Scott Hayes

Reviewer #2: No

Reviewer #3: No

---

## [Author Response · Author response to Decision Letter 0]

2 Jun 2020

Dear Editor;

Original research project; included 4 different stress treatments including; TM, DTT, 100mM and 400mM Salt stress treatments. Bioinformatics analysis of these stress treatments (including all stress) were evaluated all together.

We presented the manuscript including TM and 400mM salt stress data, since only TM and 400mM salt treatment stress gave significant results in all different stress applications and DTT and 100mM salt stress treatment did not yield significant results. However, since it is thought that TM and 400mM salt stress analysis, which are not included in the manuscript (due to the lack of meaningful results) performed with DDT-100mM Salt stress, will affect the results related to these stresses.

In this revised manuscript, all of the bioinformatics analysis has been revised so that only TM-400mM Salt stres is specific to stress. Therefore, some genes (Probe set ID), gene numbers, and gene expressions (fold change) have been changed and these changes have been updated in the manuscript. In addition, the 'Microarray data processing and analysis' part of the method has been updated according to the information such as the tool and web server used in the new analysis.

We apologize for the bioinformatic analysis of stress data that could not be included in the manuscript during the first submission.

We thank the reviewers for their careful read and thoughtful comments on previous draft. We have carefully taken their comments into consideration in preparing our revision, which has resulted in a paper that is clearer, more compelling, and broader. The following summarizes how we responded to reviewer comments. The responses are shown in bold.

Sincerely 

Ali ERGÜL

Journal Requirements:

We note that you are reporting an analysis of a microarray, next-generation sequencing, or deep sequencing data set. PLOS requires that authors comply with field-specific standards for preparation, recording, and deposition of data in repositories appropriate to their field. Please upload these data to a stable, public repository (such as ArrayExpress, Gene Expression Omnibus (GEO), DNA Data Bank of Japan (DDBJ), NCBI GenBank, NCBI Sequence Read Archive, or EMBL Nucleotide Sequence Database (ENA)). In your revised cover letter, please provide the relevant accession numbers that may be used to access these data. 

Response: All microarray data (cell files, metadata and matrix Template) was submitted to GEO (Gene Expression Omnibus, (http://www.ncbi.nlm.nih.gov/projects/geo/.) database (GEO ID: GSE150581) using NCBL (National Center for Biotechnology Information) web server.

Reviewer #1:

PONE-D-20-03767- Review Scott Hayes

Salt stress induces endoplasmic reticulum stress-responsive genes in a grapevine rootstock

The authors present their work comparing the transcriptomes of salt and Tm treated grapevine. This data-set has the potential to inform future work and may prove a valuable contribution to the field. On the whole, the author’s discussion of their results is well measured and appropriate for the level of evidence they provide. I do however feel that the manuscript could be improved if the authors were to provide additional information on the experimental set-up. In addition, communication of their results could be improved through changes in the way they present their data. 

One major point (and a pet peeve of mine!) is that at several occasions in the methods, readers are referred to the methods of other manuscripts. Not only does this waste a lot of the interested reader’s time, it may result in misunderstandings about how the work was actually performed. For example, the authors state that they performed RNA extraction from leaf samples as described in reference 52. As reference 52 is a comparison of 15 different RNA extraction methods, it is presently unclear which method they are referring to! In the interest of improving the reproducibility of these results, please include the full methods for this manuscript. 

It would be useful if the figures and their associated legends contained more information.

Response: We appreciate the Reviewer #1 suggestions and improved the full methods in details in RNA extraction in page 5 and we introduced more information in figure legends.

“Fig 1. Stem water potential of leaves treated with salt and ER stresses. Data are means of eight measurements”

- Which leaves? 

- How old were the plants?

- How were they grown?

- At what time did the treatment start?

Some of this information is in the method but having this basic information in the figure legend would help the reader to interpret the results. Additionally, the authors represent the mean of 8 measurements and so it would be good to include error bars in the figure to demonstrate the variance in the data. The authors also imply that they have performed statistical analysis of this data, but it is not clear from the figure legend what this entails.

Response: We thank the reviewer’s comments. We improved the legend and changed to “Plantlets were grown in a growth chamber for 10 weeks at 25�C. Plantlets with 4–5 leaves and 2–4 roots were chosen for use in stress treatments. Measurements were performed for ER and salt stressed samples and controls at indicated time points. Results were statistically analyzed using t-tests (p <0.05). Data are means of eight measurements”.

“Fig 2. Venn diagram for transcripts regulated at 6 h (A) and 24 h (B) by salt and ER stresses. Light green or blue represent transcripts with increased or decreased expression under salt stress, respectively. Yellow or green represent transcripts with increased or decreased expression under TM treatment”

- Personally I found this figure legend quite difficult to understand at first. I believe that the authors are comparing transcript abundance between the control conditions and NaCl or TM treatments but this is not stated in the legend.

- In my opinion, the clarity of the legend could be improved by separating out the clauses in the sentences. For example, “Microarrays were performed on control, NaCl and TM treated plants at 6 h (A) and 24 h (B). Green circles show transcripts that were at a higher abundance in NaCl than the control, and blue circles show transcripts that were at a lower abundance in NaCl than the control. Red circles show…..”

- Also I would like to request that authors use a different colour scheme, as to me, the difference between green, light green and blue is not very clear. 

Response: We reformulated the phrase and changed to “ Venn diagram of microarrays analysis performed on control, NaCl and TM treated plants at 6 h (A) and 24 h (B). Green circles show transcripts that were at a higher abundance in NaCl treated plants than the control, and blue circles show transcripts that were at a lower abundance in NaCl treated plants than the control. Yellow circles show transcripts that were at a higher abundance in TM treated plants than the control, and red circles show transcripts that were at a lower abundance in TM treated plants than the control “

We also changed the light green circle to red circles as recommended.

“Fig 4. Gene Ontology classification of salt-stress responsive genes. (A) indicates ’biological process’ and (B) indicates ‘molecular functions’ (C) indicates ‘cellular component’ for transcripts with increased or decreased expression under salt stress. Each bar represents the number of differentially expressed transcripts in each category” This figure contains so much data that it becomes difficult to see what it represents. It may be worth investigating other methods to display this data, such as a heat map for example. 

Response: Thank you for reviewer suggestions. We performed an enrichment analysis of the expression data and metabolic pathways affected by salt and ER-stress or both of them in order to make it easier to understand the data. We also rearranged Gene Ontology classification figures.

“Figure 5. Fig 5. Expression analysis of chosen genes by qRT-PCR. Total RNA was isolated from leaves of grapevine rootstock treated with NaCl (A, B) or tunicamycin (TM) (C, D) for 2. 6. 24. and 48. hours indicated times. The bars represent the relative intensity of labelled qRT-PCR products from three independent biological replicates. eIF4α was used as an internal control.”

- It is not clear to me what the difference is between A & B, and C & D. 

- Could the authors clarify what is meant by biological repeat? In many of their samples the variation between repeats is so small that error bars are not visible. In my experience, such small error bars are often seen for technical repeats (showing pipetting error from a single sample) than biological repeats (showing the difference in expression between completely separate experiments). 

- The data also appears to have been normalised relative to Ctr. If so, this should be stated. 

- If the purpose of this figure is to compare the expression of genes between TM and NaCl treatments it may be more useful to incorporate this information in a single graph for each gene. At the very least, the separate graphs could be rearranged so that they occupy the same order between the two treatments (e.g. both the top left figures showing stilbene synthase, and to their left dehydrin etc).

Response: Thanks, the reviewer for comments. The reason we put these real time PCR data is that we could validate the results of Microarray analysis data. In these figures, we are not comparing the real time PCR results of both stresses. However, in most cases, the expression of some of them are regulated by both stresses.

We also included error bars corresponding to biological repeats not the technical repeats.

As the reviewer commented on it we normalized the data relative to control and we stated in the legend. The name of the genes and their accession number are useful for readers to be able to compare to microarray data. We can also find commonly regulated genes in Table 2 for information.

Subheadings such as “Effects of salt and ER stress on cell wall and fatty acid composition” and “Effects of salt and ER stress on chromatin structure” should be changed. Something like “The effect of salt and TM on the expression of genes that are involved in cell wall modification” is much less likely to give the reader the impression that cell wall composition or chromatin structure were experimentally evaluated in this manuscript. There are several examples of these headings in the discussion that could be improved.

Response: We thank the reviewer for suggestions. We changed the headings to “Effects of salt and ER stress on the expression of genes involved in chromatin structure” and “Effects of salt and ER stress on the expression of genes involved in cell wall modification”. In discussion section we modified the headings to “Effects of salt and ER stress on the expression of genes involved in osmotic regulation” and “Effects of salt and ER stress on the expression of genes involved in carbon metabolism” and “Effects of salt and ER stress on the expression of genes involved in lipid metabolism and ROS detoxification”.

I am also interested in some technical aspect of the experimental set-up. The authors submerge the roots of their pants in liquid media containing NaCl or Tm and then take samples of the leaves at 6 and 24hs thereafter. My question is: is there is any data on the speed at which Tm travels through the plant? The relatively low number of genes affected in the leaves is surprising, given the huge effect that Tm has on stem water potential. 

An alternative hypothesis may be that Tm and NaCl both damage the root (leading to the drop in water potential and possible long-distance stress signal to the leaves). NaCl may then additionally travel to the leaves to elicit the large change in gene expression seen in this tissue. Would the authors care to comment on this?

Response: We appreciate the interest of reviewer on the subject and questions. To our knowledge there is no data at what speed TM travels through the plants. But we know that in many plants including Arabidopsis (Kamauchi et al., 2005*), ER stress genes are regulated from 2 hrs up to 24 hrs after TM treatment. In our experiments, we did not observe any regulation of gene within 2 hrs after TM treatment in leaves compared to salt stress where we had regulation of large number of genes. Overall we obtained the maximum number of genes regulated at 6 hrs and 24 hrs after TM and salt treatment. We would like to point out that there are a bunch of genes whose expression is organ specific (unpublished data). That is why we are not seeing a huge effect on leaves. By contrast in roots where TM and Salt contacts occur, even at 2hrs after TM treatment almost all ER stress genes are regulated which is really interesting. This data is the subject of another publication we are about to submit.

* Kamauchi, S., Nakatani, H., Nakano, C., & Urade, R. (2005). Gene expression in response to endoplasmic reticulum stress in Arabidopsis thaliana. The FEBS journal, 272(13), 3461-3476.

Minor comments:

Page 2: 

- “salt stress can be avoided by using salt-tolerant grape rootstocks obtained from wild Vitis

Species”. Please add a reference here.

Response: We added the reference to the phrase.

Page 3:

-Some aspects of the introduction could contain more information to enhance the readability of the MS. For example, the salt sensitivity of slp and bZip17 mutants is discussed but it is assumed that the reader already knows the function of these genes. Similarly, instead of “IRE1 protein is highly conserved in eukaryotes including plants”, something along the lines of “IRE1 is an ER-localised transmembrane kinase that is well conserved in eukaryotes” may be more informative to readers from outside the field of ER stress. As is features heavily in the presented work, it would also be useful to state in the introduction what tunicamycin is and give a brief description of how it works.

Response: We appreciate the reviewer suggestions. We introduced the explanation to the abbreviations for slp and bZip17 or bZIP60 mutants in the introduction section. Additionally, we shortened the introduction by removing certain parts related to yeast IRE1 protein which makes it heavier to understand.

Methods:

“As revealed by preliminary experiments, a quantity of 400 mM salt was sufficient for

decreasing stem water potentials while not harming the plants exposed to salt stress for the indicated times”. Please provide more details here.

Response: We performed different concentration of NaCl from 120 mM to 400 mM for stem water potential experiments. Most of the cases we can go up to 250 mM for salt stress but in case of 1616C rootstock which is very resistant to salt (0,8 g/Kg of soil) (Galet P. 1979*) we opted to use 400 mM of NaCl to induce salt stress. 

* Galet, P. (1979). A Practical Ampelography. translated and adapted by Lucie t. Morton.

Page 7:

“The expression of 107 genes changed in response to ER stress induced by TM (Fig 2A and 2B).” 

This statement is incorrect as there are several overlapping genes between the two time points, meaning that the number of genes affected is lower than 107. The same holds for similar statements made regarding salt-regulated genes on page 8.

Response: We thank the reviewer’s comments. In this paragraph, we did not mentioned the transcript whose expression unique to ER or salt stresses. On page 6 , we gave information about the number of regulated genes unique to ER or Salt stresses.

Page 18:

“…the transcript expression of an auxin binding protein ABP19a-like, which is involved in auxin signal perception,”. Could the authors please provide a reference here. 

Response: We added the reference.

Reviewer #2: 

Dear Authors, the experiment is interesting and therefore the article should be improved technically by shortening of reference list at least by 30% (in Reviewer opinion the list is too long) as well as the text. 

Response: We shortened the manuscript as much as possible.

The text needs some reorganizations as well. At present, the Results contains some parts of Methods and Discussion contains some part of Results. 

Response: We removed method parts from Results and tried to remove the results from discussion section.

Also for the Readers, it would be easier to follow the text when it is connecting reasons with effects by sentence construction using words i.e: because, therefore, for the reason, as a result. At present, the text is a set of separate paragraphs. 

Response: We gathered the seperate paragraphes as recommended.

And paragraphs, especially in Discussion are lists of examples without or with weak explanations/discussion. So please make the text shorter, clearly organized into sections and fluent.

Response: We made the text shorter including the discussion section by removing unnecessary examples.

Reviewer #3: 

The authors compared the effect of salt and tunicamycin-induced ER stress on transcriptome pattern in grapevine rootstock. The experimental system is interesting and appropriate to determine the regulatory mechanism specific either to salt or ER stress or those ones which are involved in the response to both stresses. However, the manuscript is not well-written since a large part is descriptive and the various paragraphs are not connected to each other. It is also too long, and it could be reduced by 30%.

Response: We shortened the manuscript as much as possible.

It would be interesting to check which target genes of the stress-induced transcription factors were also affected. 

Response: We thank the reviewer’s comments. It would be interesting to check target genes but as this study is a microarray research, only genes which expression varies in stress applications (ER and salt) are emphasized.

Based on this analysis, a model of regulatory network with specific and overlapping parts could be created. An enrichment analysis of the expression data could show which regulatory and metabolic pathways were mainly affected by salt and ER-stress or both of them.

Response: For pathway analysis, Probe IDs of genes with expression differences were used to download relevant cDNA sequences from the ensembl plant database (https://plants.ensembl.org/biomart/martview). KEGG (Kyoto Encyclopedia of Genes and Genomes) is a database used to understand the functions and utilities of the biological system for each stress (https://www.genome.jp/kaas-bin/kaas_main).

 The abstract should not only list the genes, but try to explaine some regulatory mechanisms. 

Response: We added KEGG pathway analysis results (as ‘KEGG pathway analysis of differentially expressed genes for both ER and salt stress were divided into four main categories including; carbohydrate metabolism, amino acid metabolism, signal transduction and lipid metabolism ’) to the abstract. 

There are some very short paragraphs (1-2 sentences) in the introduction. Longer ones should be created which allow also their better connection.

Response: We gathered the seperate paragraphes as recommended.

 Only the most interesting results should be described in the text and the greatest differences and similarities between the two types of stress should be emphasized.

Response: We kept the most interesting parts in the text.

 In the discussion the possible regulatory mechanism should be explained based on the present observations at transcriptome level and those ones available in the literature at proteome or metabolome level. Its certain parts are repetition of the results.

Response: We made the discussion shorter by removing repetitive parts.

Remarks:

1. Abstract – salt stress influences ER stress based on genes induced by both of them. The sentence should be rewritten more carefully since this statement was not proved.

Response: We removed the sentence from abstract.

2. Why 5 µg/ml tunicamycine concentration was chosen? It is possible that it was too low since only few genes were affected compared to salt stress.

Response: In many cell types, ER stress can be induced by 2.5–5 μg/mL tunicamycin (TM) (Oslowski & Urano, 20111). In plants, the concentration may vary from 0.5 µg/mL to 5 µg/ml TM (Guan et. al., 20182). In liquid medium, similar to our study, 5 µg/ml TM could be used (Liu et al., 20073; Lu et al., 20124; Cho & Kanehara, 20175; Guan et al., 20182). We would like to point out that there are a bunch of genes whose expression is organ specific (unpublished data). That is why we are not seeing a huge effect on leaves. By contrast In roots where TM and Salt contacts occur, even at 2hrs after TM treatment almost all ER stress genes are regulated which is really interesting. This data is the subject of another publication we are about to submit.

1. Oslowski, C. M., & Urano, F. (2011). Measuring ER stress and the unfolded protein response using mammalian tissue culture system. In Methods in enzymology (Vol. 490, pp. 71-92). Academic Press.

2. Guan, P., Wang, J., Li, H., Xie, C., Zhang, S., Wu, C., ... & Zheng, C. (2018). Sensitive to SALT1, an endoplasmic reticulum-localized chaperone, positively regulates salt resistance. Plant physiology, 178(3), 1390-1405.

3. Liu, J. X., Srivastava, R., Che, P., & Howell, S. H. (2007). An endoplasmic reticulum stress response in Arabidopsis is mediated by proteolytic processing and nuclear relocation of a membrane-associated transcription factor, bZIP28. The Plant Cell, 19(12), 4111-4119.

4. Lu, S. J., Yang, Z. T., Sun, L., Sun, L., Song, Z. T., & Liu, J. X. (2012). Conservation of IRE1-regulated bZIP74 mRNA unconventional splicing in rice (Oryza sativa L.) involved in ER stress responses. Molecular plant, 5(2), 504-514.

5. Cho, Y., & Kanehara, K. (2017). Endoplasmic reticulum stress response in Arabidopsis roots. Frontiers in plant science, 8, 144.

3. The length of treatments should be described in the method part.

Response: We modified the method section especially RNA extraction subheading with more details.

4. The availability of the microarray data is not given.

Response: The Microarray data obtained in the study provides findings for the investigation of the mechanism of the ER pathway at the transcriptome level and determination of the transcript candidates in response to the grapevine salt stress. These data can be used in studies aimed at determining the genetic characterization and functions of these transcripts, especially in the ER pathway.

5. In the second paragraph of the results two numbers for differentially expressed genes are mentioned but the selection criteria is not given for the first one.

Response: We modified the second paragraph of the results as ‘We used the GeneChip® Vitis vinifera Genome Array which provides comprehensive coverage of the V. vinifera (grape) genome to identify significantly differentially expressed (at least two) transcripts (p <0.01) between salt-stressed and control plants or ER-stressed and control plants by microarray analysis’.

6. The correlation between microarray and qRT-PCR data should be calculated.

Response: Validation of the microarray data by qPCR for genes whose expression was significantly correlated with microarray data in salt-stressed plants was presented in revised manuscript.

7. The activation of Pro synthesis is mentioned during the investigated stresses, but the related genes (P5CR, P5CS) are not listed among the affected ones.

Response: When the expression of Pro synthesis genes such as (P5CR, P5CS) are taken into consideration in therms of regulation, only one pyrroline-5-carboxylate synthetase (P5CS) (1619565_at) gene expression was slightly upregulated in response to both ER (2.57 fold change) and Salt (2.60 fold change) stresses (Supplemental Table 2 and 4). We added this gene to Table 2 (as a stress responsive gene).

8. The typing and grammatical errors should be corrected.

Response: We corrected some typing and grammatical errors.

---

## [Decision Letter · Decision Letter 1]

1 Jul 2020

PONE-D-20-03767R1

Salt stress induces endoplasmic reticulum stress-responsive genes in a grapevine rootstock

PLOS ONE

Dear Dr. Ali ERGÜL,

Thank you for submitting your manuscript to PLOS ONE. After careful consideration, we feel that it has merit but does not fully meet PLOS ONE’s publication criteria as it currently stands. Therefore, we invite you to submit a revised version of the manuscript that addresses the points raised during the review process.

One of the reviewers has requested for some minor formatting corrections. The authors need to address those points before getting final acceptance.

We look forward to receiving your revised manuscript.

Kind regards,

Mayank Gururani

Academic Editor

PLOS ONE

Reviewers' comments:

Reviewer's Responses to Questions

**Comments to the Author**

1. If the authors have adequately addressed your comments raised in a previous round of review and you feel that this manuscript is now acceptable for publication, you may indicate that here to bypass the “Comments to the Author” section, enter your conflict of interest statement in the “Confidential to Editor” section, and submit your "Accept" recommendation.

Reviewer #1: All comments have been addressed

Reviewer #2: All comments have been addressed

Reviewer #3: All comments have been addressed

2. Is the manuscript technically sound, and do the data support the conclusions?

Reviewer #1: Yes

Reviewer #2: Yes

Reviewer #3: Yes

3. Has the statistical analysis been performed appropriately and rigorously? 

Reviewer #1: Yes

Reviewer #2: Yes

Reviewer #3: Yes

4. Have the authors made all data underlying the findings in their manuscript fully available?

Reviewer #1: Yes

Reviewer #2: Yes

Reviewer #3: Yes

5. Is the manuscript presented in an intelligible fashion and written in standard English?

Reviewer #1: Yes

Reviewer #2: Yes

Reviewer #3: Yes

6. Review Comments to the Author

Reviewer #1: The authors present a much improved manuscript. The manuscript provides a very thorough description of gene expression changes in grapevine leaves after TM and NaCl treatment. I believe the MS now conforms to PLoS ONE’s criteria for publication as my previous concerns about the fullness of the methods has now been addressed. I believe the MS could still be slightly improved by an explanation of tunicamycin in the introduction. Currently the first mention of tunicamycin is “These mutants accumulate misfolded proteins and induce the UPR in response to tunicamycin”. It would be useful to state something along the lines of “tunicamycin blocks glycoprotein synthesis and triggers the UPR” before this statement.

Reviewer #2: Comment to the Editor and Authors:

the text of the paper was improved. The figures have been changed and are impressive. Number of literature positions was shortened.

The number of paragraphs should be reduced more, by typing continuously without line breaks (in most cases). This is only the technical issue. Division into paragraphs should be done according to the main topic of particular text fragments

I recommend the paper for publication.

Krystyna Rybka

Reviewer #3: The authors made all the recomended changes during the revision of the manuscript or they gave an appropriate answer for the remarks.

7. PLOS authors have the option to publish the peer review history of their article (what does this mean?). If published, this will include your full peer review and any attached files.

Reviewer #1: **Yes: **Scott Hayes

Reviewer #2: **Yes: **Krystyna Rybka

Reviewer #3: No

---

## [Author Response · Author response to Decision Letter 1]

5 Jul 2020

PONE-D-20-03767R1

Salt stress induces endoplasmic reticulum stress-responsive genes in a grapevine rootstock

PLOS ONE

Dear Dr. Ali ERGÜL,

Thank you for submitting your manuscript to PLOS ONE. After careful consideration, we feel that it has merit but does not fully meet PLOS ONE’s publication criteria as it currently stands. Therefore, we invite you to submit a revised version of the manuscript that addresses the points raised during the review process.

One of the reviewers has requested for some minor formatting corrections. The authors need to address those points before getting final acceptance.

We look forward to receiving your revised manuscript.

Kind regards,

Mayank Gururani

Academic Editor

PLOS ONE

Dear Editor,

Thank you and the journal reviewers for useful comments on our manuscript. As described in detail below and also on the re-submitted manuscript, we have now revised the manuscript by carefully considering these comments. 

We hope that the manuscript is now suitable for publication in PLOS ONE

Yours Sincerely

Prof. Dr. Ali ERGÜL

6. Review Comments to the Author

Reviewer #1: The authors present a much improved manuscript. The manuscript provides a very thorough description of gene expression changes in grapevine leaves after TM and NaCl treatment. I believe the MS now conforms to PLoS ONE’s criteria for publication as my previous concerns about the fullness of the methods has now been addressed. I believe the MS could still be slightly improved by an explanation of tunicamycin in the introduction. Currently the first mention of tunicamycin is “These mutants accumulate misfolded proteins and induce the UPR in response to tunicamycin”. It would be useful to state something along the lines of “tunicamycin blocks glycoprotein synthesis and triggers the UPR” before this statement.

Response: “In the revised manuscript, the statement has now been modified as per reviewer suggestion as ; Tunicamycin blocks glycoprotein synthesis and triggers the UPR. These mutants accumulate misfolded proteins and induce the UPR in response to tunicamycin.

Reviewer #2: Comment to the Editor and Authors:

the text of the paper was improved. The figures have been changed and are impressive. Number of literature positions was shortened.

The number of paragraphs should be reduced more, by typing continuously without line breaks (in most cases). This is only the technical issue. Division into paragraphs should be done according to the main topic of particular text fragments

I recommend the paper for publication.

Krystyna Rybka

Response: As per reviewer suggestion, the number of paragraphs has now reduced, by typing continuously without line breaks (in most cases). The division into paragraphs has been done according to the main topic of particular text fragments .

Reviewer #3: The authors made all the recomended changes during the revision of the manuscript or they gave an appropriate answer for the remarks

---

## [Editor Report · Decision Letter 2]

8 Jul 2020

Salt stress induces endoplasmic reticulum stress-responsive genes in a grapevine rootstock

PONE-D-20-03767R2

Dear Dr. Ali ERGÜL,

We’re pleased to inform you that your manuscript has been judged scientifically suitable for publication and will be formally accepted for publication once it meets all outstanding technical requirements.

Kind regards,

Mayank Gururani

Academic Editor

PLOS ONE
---

## [Editor Report · Acceptance letter]

16 Jul 2020

PONE-D-20-03767R2 

Salt stress induces endoplasmic reticulum stress-responsive genes in a grapevine rootstock 

Dear Dr. ERGÜL:

I'm pleased to inform you that your manuscript has been deemed suitable for publication in PLOS ONE. Congratulations! Your manuscript is now with our production department. 

Kind regards, 

on behalf of

Dr. Mayank Gururani 

Academic Editor

PLOS ONE